# Using Non-Pharmaceutical Interventions and High Isolation of Asymptomatic Carriers to Contain the Spread of SARS-CoV-2 in Nursing Homes

**DOI:** 10.3390/life12020180

**Published:** 2022-01-26

**Authors:** Alec J. Schmidt, Yury García, Diego Pinheiro, Thomas A. Reichert, Miriam Nuño

**Affiliations:** 1Deparment of Public Health Sciences, University of California Davis, Davis, CA 95616, USA; 2Centro de Investigación en Matemática Pura y Aplicada (CIMPA), University of Costa Rica, San José 11501, Costa Rica; yury.garciapuerta@ucr.ac.cr; 3Department of Internal Medicine, School of Medicine, University of California Davis, Sacramento, CA 95817, USA; diego.silva@unicap.br; 4Entropy Research Institute, Lincoln, MA 01773, USA; treichert@entropylimited.com

**Keywords:** non-pharmaceutical interventions, nursing homes, COVID-19, silent transmission, pre-symptomatic carriers

## Abstract

More than 40% of the deaths recorded in the first wave of the SARS-CoV-2 pandemic were linked to nursing homes. Not only are the residents of long-term care facilities (LTCFs) typically older and more susceptible to endemic infections, the facilities’ high degree of connection to wider communities makes them especially vulnerable to local COVID-19 outbreaks. In 2008, in the wake of the SARS-CoV-1 and MERS epidemics and anticipating an influenza pandemic, we created a stochastic compartmental model to evaluate the deployment of non-pharmaceutical interventions (NPIs) in LTCFs during influenza epidemics. In that model, the most effective NPI by far was a staff schedule consisting of 5-day duty periods with onsite residence, followed by an 4-to-5 day off-duty period with a 3-day quarantine period just prior to the return to work. Unlike influenza, COVID-19 appears to have significant rates of pre-symptomatic transmission. In this study, we modified our prior modeling framework to include new parameters and a set of NPIs to identify and control the degree of pre-symptomatic transmission. We found that infections, deaths, hospitalizations, and ICU utilization were projected to be high and largely irreducible, even with rigorous application of all defined NPIs, unless pre-symptomatic carriers can be identified and isolated at high rates. We found that increasingly rigorous application of NPIs is likely to significantly decrease the peak of infections; but even with complete isolation of symptomatic persons, and a 50% reduction in silent transmission, the attack rate is projected to be nearly 95%.

## 1. Introduction

Despite being less than 1% of the population of the United States, 8% of all confirmed cases and more than 41% of recorded COVID-19 deaths during the first wave of infections in US were linked to nursing homes and other long-term care facilities (LTCFs). The EU and Republic of Korea saw a similar skew of cases in early waves [1,2]. Caring for our aging population has seldom been a top priority [3,4], even though the proportion of US elderly residing in LTCFs is greater than anywhere else in the world. Nursing homes and residential care communities staffed by more than 1 million nurses, aides, and social workers are home to more than 2.2 million residents, most of advanced age [5], and thus at a higher risk of infection and mortality from COVID-19 [6,7,8]. In this population, the prevalence of advanced age and significant comorbidities such as diabetes and hypertension [9,10,11,12,13] substantially increases the risk of hospitalization, severe symptoms, and death.

Since the 1918 influenza pandemic there has been extensive discussion about how to best prepare for another pandemic [14,15,16,17,18], with some special considerations for LTCF residents [19,20,21,22]. In 2018, the World Health Organization (WHO) released the Global Influenza Strategy 2019–2030 with the goals of reducing the burden of seasonal influenza, minimizing the risk of zoonotic influenza, and mitigating the effects of pandemic influenza. These efforts, however, have provided limited guidance on the implementation of non-pharmaceutical interventions (NPIs) for controlling the spread of pandemic infection [18]. Controlling exposure pathways through social distancing measures, personal protection measures, and community-based interventions is critically important, uniquely so during the period of time between pandemic pathogen emergence and the development and distribution of therapeutics and vaccines, and during periods of uncertainty when a new variant with breakthrough potential emerges.

In 2008, motivated by mounting concern about pandemic influenza, we produced a mathematical model that anticipated some of the challenges that nursing homes would face in the event of a new influenza pandemic prior to vaccine deployment. The most effective modeled plan required virtually complete facility isolation, complete visitation restriction, rotating staff schedules with a variable isolation period prior to employee reentry, and stringent mitigation protocols imposed on all high-priority services entering a facility [22]. Without access to vaccines or other therapeutics, we concluded that the effectiveness of such plans mostly depended on preventing staff from introducing an infection acquired outside of the facility, as mitigation after introduction is rarely effective. The successful containment of SARS/MERS and the relatively low level virulence of the 2009 H1N1 influenza virus made the considerable investment and social disruption required for such a pandemic plan difficult to justify.

COVID-19 has dramatically changed that calculation. Though some countries in the Global North currently have high vaccine coverage in assisted living facilities, immune protection from the first round of vaccines is demonstrably waning, especially during the surge of the Omicron variant in the winter of 2022, while global inequities in distribution have left some countries unable to implement widespread vaccination altogether. Thus, exploring the effectiveness of non-pharmaceutical interventions (NPIs) in LTCFs remains as relevant now as it was at the beginning of the pandemic. Given the unlikely global availability of vaccines and chronic control deficits, it is critical to identify opportunities and mechanisms that reduce the risk of introducing or re-introducing COVID-19 into LTCFs.

Influenza viruses are epidemiologically distinct from the SARS-CoV-2 virus. While the incubation period of influenza viruses is typically 2–3 days, that of COVID-19 appears to be between 5 and 9 days, with a small number of cases taking 12+ days to manifest symptoms [23,24,25,26,27]. Our models for influenza recommended the isolation of staff for 3+ days at the end of off-duty periods to allow symptoms to manifest after exposure. However, the longer and uncertain incubation period of COVID-19 would require staff to quarantine for up to 2 weeks at the end of an off-duty period for conservative measures of effectiveness. Even one week is a non-starter, as continuity of care is an important concept in these skilled nursing facilities [28]. Additionally, serial interval studies provide mounting evidence of a period of pre-symptomatic infectiousness of multiple days, during which an infected individual can unknowingly expose others [29,30,31,32]. The pre-symptomatic transmission has likely contributed significantly to COVID-19 spread in many LTCFs around the world and continues to be a significant challenge that NPIs based on influenza transmission dynamics, or even SARS-CoV-1, do not address [33,34,35,36,37].

In this study, we adapted our previous stochastic SEIR model of a pandemic influenza outbreak in a nursing home to reflect the longer incubation period and significant pre-symptomatic transmission risk of COVID-19. We then characterized the performance of four distinct NPI scenarios across a range of reproductive numbers and isolation rates for pre-symptomatic individuals to identify approaches that most reduced the attack rate as well as the timing and magnitude of the epidemic peak.

## 2. Materials and Methods

Nuño et al. 2008 [22] analyzed how effectively four different NPI scenarios prevented the introduction of infection to a nursing home resident population during a hypothetical local outbreak of pandemic influenza. We adapted the stochastic, compartmental Susceptible-Exposed-Infected-Recovered (SEIR) model from this previous study to reflect our current knowledge of progression and transmission parameters for SARS-CoV-2, including significant modifications for residents, to model outcomes in a nursing home with 200 susceptible residents and 83 total staff. Multiple NPI scenarios with varying levels of rigor were applied across incremental pre-symptomatic isolation rates (indicating differing levels of asymptomatic testing access) and viable values for $R_{0}$ to see how the attack rate, mortality rate, number of hospitalizations, and number of ICU admissions responded. Each intervention category described by the original model was restructured and re-parameterized to reflect the current suggested NPI policies for nursing homes [6,28,38]. NPIs modeled include isolation of symptomatic residents and barring symptomatic staff from work; staff and visitor use of PPE when interacting with residents; temperature monitoring for incoming staff and visitors; staff schedules that account for incubation and isolation periods; and restrictions on visitor entry. Continuous outcomes among NPI scenarios were assessed for significance by calculating the difference between two scenarios and applying one-sided t-tests with a null hypothesis set to a difference of 0.

### 2.1. Mathematical Model

Four main actors were considered: residents, who could only be exposed in the facility and move to the hospital or ICU; staff, who could be exposed in the community or in the facility; visitors, who had limitations on visiting hours, and could be exposed in the community or the facility; and community members, within which the local outbreak begins. All actors within the facility at a given time were assumed to be mixing randomly, as were all actors in the community. Though permanence of immunity is still unknown, we assume that recovery provides complete immunity for the remainder of the 200-day simulation. A detailed accounting of the structure and parameterization of the model can be found in the Supporting Information.

In the SEIR model, where i,j are the actor type (resident, staff, visitor, community member) and their location (facility or community) respectively, all who can be infected begin in the susceptible compartment $S_{ij}$. When successfully exposed, they move to the asymptomatic/pre-symptomatic silent infection compartment, $E_{ij}$, where they can infect other susceptible actors. Some develop symptoms, moving to the infected compartment $I_{ij}$, while others do not and ultimately recover and are removed from the modeling population (complete immunity). Since hospital and ICU capacity are critical concerns in the COVID-19 pandemic, we added two new compartments to the resident outbreak model: infected residents who required hospitalization move to the hospital compartment, $H_{R}$, and residents who require intensive care move to the ICU compartment, $U_{R}$. Residents could suffer a mortality event in both $H_{R}$ and $U_{R}$, which would remove them from the modeling population. The parameters and definitions presented in Table 1 reflect the known characteristics of the SARS-CoV-2 virus for the first wave of the pandemic.

A full schematic of the model is available in supporting materials (Appendix A). State transition rates for residents in the facility are highlighted in Equation (Equation 1), below. Exposure rates were related to the infection force, λ, which modifies the transmission rate and transmission reduction from protective measures by taking transient actors into account (Equation (Equation 2)). Specifications of β, which relates to $R_{0}$, are available in Appendix A.

### 2.2. Baseline Scenario

The Baseline scenario describes the case where no specific preventative action is taken. Symptomatic residents are not isolated, symptomatic staff are allowed to report to work, staff and visitors are not required to use PPE in normal interactions with residents, temperature and other symptom checks are not required for entry, visitors are allowed two hours of visitation time per week, and staff have a normal 5 days on/2 days off work week.
(1)$$\begin{array}{cc} \frac{dS_{R}}{dt} & =-\lambda_{in}S_{R}\\ \frac{dE_{R}}{dt} & =\lambda_{in}S_{R}-\left(1-m\right)\phi_{R}E_{R}-m\phi_{A}E_{R}\\ \frac{dA_{R}}{dt} & =\left(1-m\right)\phi_{R}E_{R}-\gamma_{A}A_{R}\\ \frac{dI_{R}}{dt} & =m\phi_{A}E_{R}-\gamma_{I}I_{R}-\theta_{1}I_{R}\\ \frac{dH_{R}}{dt} & =\theta_{1}I_{R}-\left(\gamma_{H}+\mu_{H}\right)H_{R}-\theta_{2}H_{R}\\ \frac{dU_{R}}{dt} & =\theta_{2}H_{R}-\left(\gamma_{U}+\mu_{U}\right)U\\ \frac{dR_{R}}{dt} & =\gamma_{A}A_{R}+\gamma_{I}I_{R}+\gamma_{H}H_{R}+\gamma_{U}U_{R} \end{array}$$
(2)$$\lambda_{in}=\frac{\sum_{i=R,SF,VF}\beta_{i}\rho_{i}(\pi_{i}I_{i}\left(t\right)+\eta \left(A_{i}\left(t\right)+E_{i}\left(i\right)\right)}{N_{in}}$$

#### 2.2.1. NPI Categories 1–2

Categories 1–2 is defined by isolating symptomatic individuals, light mitigation of spread within the facility, and no restrictions for entrance into the facility by asymptomatic individuals. Isolation of symptomatic residents is required, and symptomatic staff are told to stay home. PPE is required for staff and visitors interacting with residents. Otherwise, entry to the facility does not require symptom checks, staff maintain their standard schedules, and visitation hours remain at baseline levels.

#### 2.2.2. NPI Categories 3–4

Categories 3–4 applies all measures in Categories 1–2 in addition to tighter restrictions on entry to the facility. Temperature checks are done for anyone entering the facility. Staff switch to 12-h shifts with 3 days off at a time to allow for more time for symptoms to develop before returning to work. Visitor hours are reduced to half of Baseline.

#### 2.2.3. NPI Category 5

Category 5 implements the most restrictive versions of the previous NPIs. Category 1–2 restrictions are still in place and temperature checks are required for entry. Staff switch to a 4 days on/4 days off schedule to further extend time away from residents. Visitors are completely disallowed. Complete specifications and parameterizations for each category are presented in Appendix A.

### 2.3. Solving the Model

The literature reports that SARS-CoV-2 has a basic reproduction number, $R_{0}$, that ranges from 1.5 to 6.47 in the early stages of an outbreak [39]. We narrowed our analysis down to an interval of values from 2.0 to 4.0 in increments of 0.2 by varying β from 0.61 to 1.21 in intervals of 0.06. Given the strong evidence that silent infections facilitate the rapid spread of the virus [48,49,50,51], we repeated each simulation at 0% to 90% (in increments of 10%) of exposed individuals being identified and isolated, simulating actors isolating themselves after contact with a known positive individual regardless of symptom status or an increasingly stringent testing regimen [52]. Since the model facility has a finite population and the outcomes of interest were based on counts, stochasticity was introduced by applying a Poisson distribution to all state transitions [53]. 100 iterations were run for each combination of inputs to quantify the uncertainty. Simulations were executed in MATLAB (vR2020a, The MathWorks, Inc., Natick, MA, USA, 2020).

### 2.4. Sensitivity Analysis

The package ODEsensitivity (v1.1.2; Weber, Theers, and Surmann, 2019) for R (v3.6.3; R Core Team 2020) was used to calculate first-order Sobol’ indices for each primary model output with respect to the two main inputs of concern, $R_{0}$ and asymptomatic isolation rate [54]. The Sobol’ method is a generalized sensitivity analysis (GSA) that provides robust variance comparisons independent of the linearity or monotonicity of a model; outputs used here are the first-order Sobol’ index, which is the ratio of a model’s variability from the parameter alone to the total variability in the full model.

## 3. Results

Four different combinations of increasingly restrictive NPIs (Baseline, Categories 1–2, Categories 3–4, and Category 5) were simulated with incremental isolation rates for asymptomatic cases (0%, 50%, 60%, 70%, 80%, and 90%), each with a $R_{0}$ ranging from 2.0 to 4.0 in intervals of 0.2. For each scenario, 8 outcomes were calculated: total resident infections and attack rate, total resident hospitalizations, total resident ICU admissions, total resident deaths and mortality rate, number of symptomatic infections at peak, and the timing of peak infections.

Figure 1 displays attack and mortality rates as well as hospital and ICU admissions observed as model outputs across $R_{0}$ values for the four NPI scenarios at 90% pre-symptomatic isolation. Similarly, Figure 2 (and Appendix A) also display attack rates but at a high-resolution intervals of 2% for pre-symptomatic isolation rates between 70% and 90%. A complete accounting of mean values and 95% confidence intervals (95% CIs) are presented in Table 2. All parenthetical ranges subsequently reported represent 95% CIs. Notably, when the pre-symptomatic isolation rate was 0% (i.e., no isolation), the four NPI scenarios were indistinguishable in terms of the total infections, hospitalizations, ICU admissions, and deaths (Appendix A).

### 3.1. Attack Rate

In the baseline scenario, total resident infections remained close to 190, or a 95% attack rate, regardless of $R_{0}$ precisely, 189.6 (188–196) at $R_{0}=2.0$ and 190.8 (189–195) at $R_{0}=4.0$. At a 0% pre-symptomatic isolation rate, no NPI scenario showed significant reductions in the attack rate (Appendix A). At 50% isolation, Categories 1–2, Categories 3–4, and Category 5 scenarios all led to lower total infections for $R_{0}<3.0$, but rapidly deteriorated for higher $R_{0}$ values (Figure 2 and Appendix A). All scenarios showed large reductions in attack rate at 90% isolation (Figure 1, Table 2, and Appendix A). Hospitalizations, ICU admissions, and mortality rate were all reduced as attack rate fell. Where differences were observable among these four scenarios, the Category 5 scenario outperformed all predecessors. Figure 2 shows that such large reductions only abruptly occurred after higher isolation rates were reached. In the worst-case scenario of $R_{0}=4.0$, for instance, reducing attack rate to 80% required isolation rates of 76% for Category 5, 80% for Categories 3–4, and 84% for Categories 1–2. The lowest attack rate achieved by our model during the worst-case scenario of $R_{0}=4.0$ was 43.2% (38.3–59.0%) at 90% isolation using Category 5 interventions. See Appendix A for the full range of calculated values.

### 3.2. Peak Infections and Peak Times

At 0% isolation, all three NPI scenarios showed significant reductions in peak symptomatic infections compared to Baseline. Across all values for $R_{0}$, peak numbers of infections were reduced by an average of 3.16 (0.76–5.57), 4.03 (1.47–6.58), and 4.28 (2.27–6.28) infections for Categories 1–2, Categories 3–4, and Category 5, respectively. When compared to baseline across all values of $R_{0}$, peak times were delayed by an average of 3.61 (1.03–6.19), 4.84 (1.33–8.36), and 9.65 (4.70–14.6) days for Categories 1–2, Categories 3–4, and Category 5, respectively. While Categories 1–2 and Categories 3–4 were not significantly different from each other, Category 5 consistently showed longer delays than the other two (*p* = 0.0004 and 0.001). See Appendix A for more details.

As isolation rates increased incrementally, while Categories 1–2 and Categories 3–4 frequently did not show significant differences from each other (α=0.05). Category 5 consistently had the largest decreases in peak size and delays to peak time at 90% isolation rate: the peak size had been reduced by an average of 52.9 (49.8–56.0) symptomatic infections, and the peak time delay increased to 32.6 (25.5–39.6) days. Except for the difference in peak time delay between 80% and 90% isolation (*p* = 0.496) the performance of every NPI Category was significantly improved over previous isolation rates (α=0.05).

### 3.3. Sensitivity Analysis

Results of the Sobol’ method sensitivity analysis can be found in Appendix A. First-order Sobol’ indices were calculated for $R_{0}$ and the asymptomatic isolation rate with respect to primary outputs. Variability in attack rate was more sensitive to the isolation rate for the first 125 days of the outbreak, with a maximum value of over 0.70 at day 27. Similarly, variability in hospitalizations was more sensitive to isolation rate for the first 155 days of the outbreak, with a maximum value of almost 0.75 before day 50. Variability in ICU admissions followed a pattern similar to hospitalizations. Variance in mortality rate, however, was more sensitive to changes in $R_{0}$ than isolation rate, except for during the early days of an outbreak where contributions were roughly equal.

## 4. Discussion

Community outbreaks were almost always followed by an outbreak in our model, indicating that even the most stringent of NPI scenarios are unlikely to prevent introduction altogether, even when $R_{0}$ was as low as 2.0, without being able to identify and isolate asymptomatic cases. Baseline conditions invariably lead to an attack rate of ≥95%, neither varying with NPI scenario nor with the pre-symptomatic isolation rate as defined. Our NPI scenarios, which were parameterized to reflect commonly endorsed control methods [6,28,38] in nursing homes and other group living situations, all failed to consistently prevent the virus from entering the facility (average attack rates significantly >0% in all simulations). Even the most restrictive intervention plan, Category 5, did not make significant differences in attack or mortality rate until 76% of asymptomatic cases were successfully isolated, where a minimal reduction in attack rate to 80% was achieved when $R_{0}$ = 4.0 (Figure 2). Thus, even the most stringent social controls we modeled were undermined by a long period of pre-symptomatic infectivity. This is consistent with assessments made on many individual nursing homes, which highlighted empirical evidence that pre-symptomatic transmission appears to be the largest contributor to their failure to control an outbreak [33,34,35,36,37].

The above is not to be taken as an argument that these NPI scenarios should be jettisoned. While they were unable to reduce the size of the outbreak without fastidious control of pre-symptomatic carriers over the 200 days of simulation, they significantly modified outbreak dynamics in terms of timing and shape of the epidemic curve (Appendix A). Even with no ability to identify or modify pre-symptomatic isolation, as would be the case where testing supplies are limited, all scenarios significantly reduced and delayed the infection peak compared to Baseline. Since each Category expands upon the previous lower-ordinal category, it is not a surprise that the model demonstrated increasing effectiveness with increasing rigor. However, it is notable that not all amplifications resulted in differences that were significant. Comparing the scenarios to each other, the quantitative difference in impact between Categories 3–4 and Category 5 was larger than that between Categories 1–2 and Categories 3–4. Absent the effects of isolation rate and $R_{0}$, longer periods of off-duty time provided to staff, greater detection of symptomatic individuals, and the blanket ban on visitors remain important contributors to mitigation. Even with an isolation rate of 0%, Category 5 bought the facility about a week of time delay for the epidemic peak. In the real-world, an extra week to act once an outbreak has started allows for additional, possibly novel reduction measures to be implemented, and reduces the immediate pressure on local hospital systems, even if the total number of hospitalizations is not reduced. NPIs, as they are currently conceived and implemented, remain important to our ability to “flatten the curve”. Quantification of such benefits is beyond the scope of this model and has been addressed elsewhere [15,55,56,57].

Inclusion of pre-symptomatic isolation rate as an input unbound to the NPI categories allowed us to explore variations in outcome independent of other NPI elements. The implementation of the scenarios we used as model inputs would likely include some inherent isolation rate in real world applications. However, the high isolation rate our model required for any of the NPI scenarios to effectively reduce attack rates would require a substantial investment of testing resources for the identification of asymptomatic individuals of sufficient sensitivity. Frequent and rapid testing for the presence of viral antigen using a technology and implementation capable of sensitivity at or above the indicated level of isolation is essential. This level of identification must be accompanied by facility and staff support to accommodate isolation periods for individuals who have had contact with known infected persons. Without the ability to implement testing and contact tracing in a way that upwards of three-quarters of potential asymptomatic, infectious carriers are continuously discovered, even current best-practices will see only small reductions on the total impact on the facility. Thus, proactive actions to isolate potential cases and identify asymptomatic cases, combined with rigorous control measures appear to be required to prevent the rapid spread of the SARS-CoV-2 virus in group settings [36,48,58].

### 4.1. Isolation Rate and Rapid Antigen Tests

Our model demonstrates the importance of planning a tailored pre-symptomatic isolation rate for the level of rigor at which other NPIs are implemented. The specificity of tests on offer appears generally quite high when implemented correctly [59]. Falsely identifying an LTCF resident or staff member as positive would be of little concern if the proper response were non-disruptive. If this information is instead used to transfer such residents to a consolidated facility such as a ward or separate building, or to shift staff schedules to allow for convalescence, then the matter becomes consequential and an engineered solution will necessarily walk a fine line between isolation based on positive tests and requiring a negative test to enter the facility. For Category 5 approaches, facilities should aim to achieve an absolute minimum of 76% isolation rate of pre-symptomatic individuals; if a less rigorous approach (lower Category) is all that is available, or a higher reduction in attack rate is desired, the burden of isolation becomes substantially higher. A perfectly sensitive test would require testing the target percentage of all person-days at the facility (followed by perfect isolation procedures); however, the four rapid antigen tests approved for use in the United States by the FDA under the Emergency Use Act (EUA) do not offer perfect sensitivity: 97.1% (85.1–99.9%; BinaxNOW COVID-19 Ag Card), 97.6% (91.6–99.3%; LumiraDx SARS-CoV-2 Ag Test), 96.7% (83.3–99.4%; Sofia SARS Antigen FIA), and 84% (67–93%; BD Veritor System) compared to PCR tests. Before rigorous evaluation of baseline measurements of success (such as positive predictive value, or PPV), a conservative sampling plan should use the lower bounds of possible sensitivities. Generally, Equation (Equation 3) displays the simple relationship between the target isolation rate $I_{Target}$, the lower bound test sensitivity $Se_{LB}$, and the proportion of person-days that testing needs to be done to achieve that rate, PPD.
(3)$$PPD=\frac{I_{Target}}{Se_{LB}}$$

If PPD=1, every resident and staff must be tested daily to reach the target isolation rate with no room for error; PPD>1 means the pre-test probability of false negatives likely prevents that test from achieving the desired pre-symptomatic isolation rate; and PPD<1 indicates that testing all person-days may not be necessary to achieve the desired isolation rate. The lower bounds of sensitivities for the available tests are displayed in Figure 2 along with a heat map of the average attack rate at different isolation rates; the region to the left of a given line highlights the isolation rates achievable with that test in a conservative plan. Note that a 76% isolation rate already rejects the use of the BD Veritor System as insufficiently sensitive, while the LumiraDx test would reliably achieve desired isolation rates up to the 90% investigated in this model. Utilizing a less rigorous approach would increase the isolation rate required to achieve the same reduction in attack rate.

### 4.2. Conclusions

These results stand in stark comparison with the results of our foundational model on pandemic influenza, which found significant reductions in impact from even Category 1–2 responses [22]. Adapting this model to COVID-19, and especially incorporating the extended pre-symptomatic period, has brought to light the requirements that all of high levels of detection, isolation of asymptomatic carriers, and rigorous applications of suggested NPIs (Category 5) are necessary for control. Results here should be taken as a demonstration of relative importance of NPIs and isolation of pre-symptomatic cases but not as predictions for absolute magnitude for real-world facilities. However, these results are consistent with empirical reports from facilities in high-risk areas, in which the absence of control measures and most particularly, merely identifying and isolating only symptomatic cases, a COVID-19 outbreak will likely spread rapidly within all long term care facilities [8,33,36,37].

It should be noted that a critical portion of a Category 5 response includes at least 4 days of time off-duty between shifts for nurses and other staff in LTCFs. Of the components that fed into our NPIs, this may also be the most difficult to implement in reality, as nursing staff shortages have plagued LTCF interventions worldwide [1,7,60,61,62]. While having highly-trained nursing staff can improve outcomes and the efficacy of prevention measures [2], many are reporting extreme levels of stress and burnout [62]. Maneuvering LTCFs into a position where they can offer nursing staff ample time off can provide a two-fold benefit: reduce stress and burnout on critical actors in the fight against outbreaks; and providing time for convalescence and the identification of cases among staff that could lead to further outbreaks.

### 4.3. Limitations

There are several limitations that are worth considering with the proposed modeling approach presented in this study. Mathematical models are deliberate simplifications of chaotic natural processes, and thus the numeric outcomes presented in this paper should not be used as hard targets for choosing which interventions to apply. The isolation rates and their relationships to model outputs can be found in Figure 2; these represent the form, if not the precise detail of real-world implementation of interventions such as these. Our model assumes that coronavirus transmission occurs from direct human-to-human contact; however, studies have shown that indirect transmission through surfaces and shared air circulation systems can also occur. While we incorporated stochastic fluctuation in the process of transmission, our model does not consider differential rates of transmission due to changes in infection control measures or individual-based variation and contact structure. Neither age nor comorbidity status was included as predictors of transmission, hospitalization, or death, so agents only differed by access to specific compartments. Thus, results should be read in terms of proportional change and not absolute values. Our mortality rates were generally lower than what has been observed in nursing home outbreaks across the US, reflecting this model’s focus on transmission dynamics and not the biological processes surround infection.

## Figures and Tables

**Figure 1 life-12-00180-f001:**
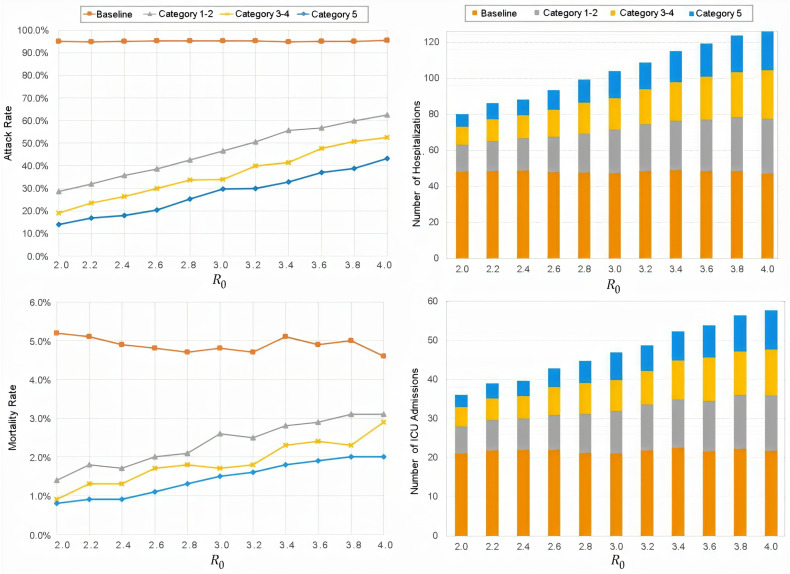
Attack rate, mortality rate, hospital and ICU admissions at 90% isolation.

**Figure 2 life-12-00180-f002:**
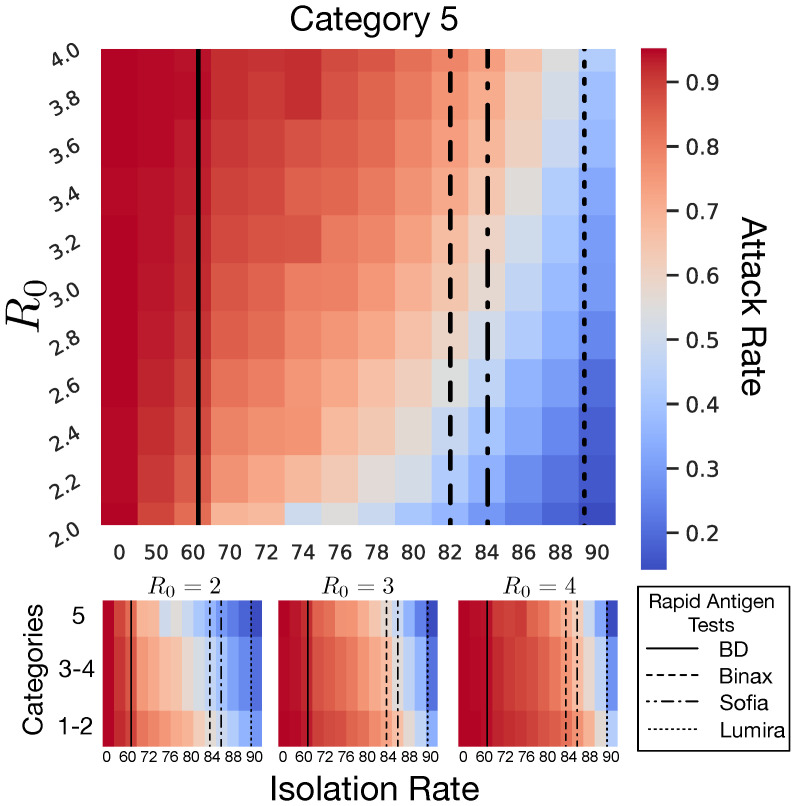
Detailed differences in attack rate for Category 5 scenarios over different isolation rates. Black lines represent the sensitivities of four rapid antigen tests approved by the FDA for detection of SARS-CoV-2 isolation rates higher than the test sensitivities are not achievable with the given test.

**Table 1 life-12-00180-t001:** Parameters and starting values for the model.

Variables	Parameters	Values	References
$R_{0}$	Basic reproduction number	2.0–4.0	[39]
$1/\phi_{i}^{+}$	Average latency period	5.0 days	[23,40,41]
$1/\gamma_{A},1/\gamma_{I}$	Average recovery period for asymptomatic/symptomatic	14 days	[42,43]
$\rho_{i}^{+}$	Transmission reduction for social use of face mask	(1–0.146)	[44,45,46,47]
$\theta_{H}$	Rate at which a symptomatic person requires hospitalization	0.16/4	[40,41]
$\theta_{U}$	Rate at which a hospitalized person requires an ICU	0.32/4	[40,41]
$1/\gamma_{H}$	Average number of days a person remains hospitalized (in ward)	1/11	[40,41]
$1/\gamma_{U}$	Average days that a person stays in ICU	1/12	[40,41]
$\mu_{H}$	Mortality rate in hospitalization	0.15/11	[40,41]
$\mu_{U}$	ICU mortality rate	0.33/12	[40,41]
Estimated			
m	Fraction of exposed that progress to infection	0.667	
$p_{A}$	Probability of having asymptomatic escape monitoring efforts	1	
$\beta_{i}^{+}$	Transmission rate		
η	Proportion of people who do not isolate themselves and continue to contribute to new infections		
Epidemiological Classes	Initial Population Size		
$S_{C}$; $S_{R}$; $S_{VC}$; $S_{VF}$; $S_{SC}$; $S_{SF}$	50,000; 200; 27; 13; 8; 75		
$E_{C}$; $E_{R}$; $E_{VC};E_{VF};E_{SC};E_{SF}$	1; 0; 1; 0; 1; 0		
$I_{C};I_{R};I_{VC};I_{VF};I_{SC};I_{SF}$	1; 0; 1; 0; 1;0		
$A_{i}^{+};R_{i}^{+}$	0; 0		
$H_{R};U_{R}$	0; 0;		
$i^{+}=R,SF,VF,SC,VC$	The subscripts *C*, *S*, and *V* are for general community, staff and visitors.		
	SF and VF correspond to staff and visitors inside the facility.		
	SC and VC are for Staff and visitors outside the nursing home (in the community).		

**Table 2 life-12-00180-t002:** Means and 95% CIs for the four main outcomes at 90% isolation.

Baseline						Category 1–2				
$R_{0}$	**Epidemic Size**	**Hospitalization**	**ICU**	**Death**	**Staff**	**Epidemic Size**	**Hospitalization**	**ICU**	**Death**	**Staff**
2.0	(188, 190, 196)	(45, 48, 61)	(18, 21, 30)	(8, 10, 17)	(25, 28, 36)	(48, 57, 81)	(11, 15, 24)	(5, 7, 13)	(2, 3, 6)	(25, 28, 36)
2.2	(187, 190, 196)	(44, 48, 64)	(19, 22, 31)	(8, 10,18)	(25, 29, 36)	(54, 64, 86)	(13, 17 25)	(6, 8, 15)	(2, 4, 7)	(26, 28, 37)
2.4	(188, 190, 196)	(44, 49, 62)	(19, 22, 30)	(8, 10, 16)	(26, 28, 36)	(61, 71, 96)	(14, 18, 28)	(6, 8, 15)	(2, 3, 8)	(25, 28, 36)
2.6	(189, 190, 195)	(44, 48, 58)	(19, 22, 31)	(8, 10, 14)	(25, 29, 39)	(69, 77, 102)	(16, 20, 32)	(6, 9, 16)	(2, 4, 9)	(26, 28, 37)
2.8	(189, 191, 195)	(43, 47, 59)	(18, 21, 30)	(7, 9, 16)	(25, 28, 38)	(77, 85, 109)	(18, 22, 33)	(7, 10, 18)	(3, 4, 10)	(25, 28, 38)
3.0	(189, 190, 196)	(44, 47, 57)	(18, 21, 33)	(8, 10, 16)	(26, 28, 36)	(82, 93, 122)	(20, 24, 34)	(8, 11, 18)	( 4, 5, 10)	(25, 28, 35)
3.2	(189, 191, 196)	(45, 48, 63)	(19, 22, 31)	(8, 9, 15)	(26, 29, 37)	(92, 101, 129)	(23, 26, 39)	(9, 12, 20)	(4, 5, 10)	(25, 28, 36)
3.4	(187, 190, 196)	(45, 49, 61)	(20, 22, 34)	(8, 10, 17)	(26, 29, 36)	(102, 111, 140)	(24, 28, 38)	(10, 12, 19)	(4, 6, 11)	(26, 29, 39)
3.6	(188, 190, 195)	(45, 48, 62)	(19, 22, 30)	(7, 10, 17)	(25, 28, 37)	(105, 113, 137)	(25, 29, 41)	(10, 13, 20)	(4, 6, 11)	(26, 29, 38)
3.8	(188, 190, 195)	(45, 49, 60)	(19, 22, 30)	(8, 10, 16)	(25, 28, 37)	(112, 120, 141)	(26, 30, 41)	(11, 14, 21)	(4, 6, 12)	(26, 29, 37)
4.0	(189, 191, 195)	(42, 47, 57)	(19, 22, 29)	(7, 9, 15)	(26, 29, 39)	(117, 125, 150)	(26, 31, 42)	(12, 14, 21)	(4, 6, 12)	(26, 29, 35)
Category 3–4						Category 5				
$R_{0}$	Epidemic Size	Hospitalization	ICU	Death	Staff	Epidemic Size	Hosp	ICU	Death	Staff
2.0	(30, 38, 62)	(7, 10, 17)	(3, 5, 10)	(1, 2, 5)	(40, 43, 54)	(22, 28, 46)	(5, 7, 13)	(2, 3, 7)	(1, 2, 4)	(39, 42, 50)
2.2	(38, 47, 71)	(9, 12, 21)	(4, 6, 12)	(1, 3, 6)	(39, 42, 49)	(24, 34, 58)	(6, 9, 18)	(2, 4, 9)	(1, 2, 5)	(39, 42, 52)
2.4	(44, 53, 84)	(10, 13, 21)	(4, 6, 11)	(1, 3, 6)	( 40, 42, 52)	(27, 36, 54)	(6, 9, 17)	(2, 4, 10)	(1, 2, 6)	(39, 42, 51)
2.6	(50, 60, 87)	(12, 15, 24)	(5, 7, 14)	(2, 3, 8)	( 40, 43, 50)	(32, 41, 71)	(8, 11, 20)	(3, 5, 10)	(1, 2, 6)	(38, 42, 51)
2.8	(57, 67, 96)	(14, 17, 27)	(5, 8, 15)	(2, 4, 9)	(40, 43, 51)	(39, 51, 80)	(8, 13, 25)	(3, 6, 14)	(1, 3, 7)	(39, 43, 51)
3.0	(58, 68, 98)	(14, 18, 28)	(6, 8, 14)	(2, 3, 8)	(40, 42, 52)	(48, 59, 97)	(11, 15, 26)	(5, 7, 15)	(2, 3, 9)	(39, 42, 51)
3.2	(69, 80, 110)	(16, 19, 30)	(7, 9, 14)	(2, 4, 8)	( 40, 43, 51)	(49, 60, 97)	(10, 15, 25)	(5, 7, 12)	(2, 3, 7)	(39, 42, 52)
3.4	(73, 83, 112)	(17, 21, 35)	(7, 10, 19)	(3, 5, 10)	(39, 43, 51)	(50, 66, 100)	(13, 17, 28)	(5, 7, 15)	(2, 4, 8)	(39, 43, 55)
3.6	(83, 95, 130)	(20, 24, 35)	(9, 11, 18)	(3, 5, 9)	(40, 43, 51)	(63, 74.2, 111)	(15, 18, 30)	(6, 8, 15)	(2, 4, 8)	(40, 43, 52)
3.8	(86, 102, 142)	(21, 25, 36)	(9, 11, 19)	(3, 5, 11)	(39, 43, 52)	(68, 78, 110)	(16, 20, 35)	(7, 9, 16)	(3, 4, 10)	(40, 42, 51)
4.0	(97, 105, 134)	(24, 27, 40)	(10, 12, 18)	(4, 6, 11)	(40, 43, 52)	(77, 86, 118)	(17, 21, 36)	(7, 10, 18)	(2, 4, 8)	(40, 43, 52)

## Data Availability

The computational system and parameters are available at the following github link https://github.com/yurygarcia26/Nursing_Home/tree/main (accessed on 8 October 2020).

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
