# Peer review of "Using Non-Pharmaceutical Interventions and High Isolation of Asymptomatic Carriers to Contain the Spread of SARS-CoV-2 in Nursing Homes"

_life, 2022, doi:10.3390/life12020180_

Round 1

Reviewer 1 Report

Dear  authors 

Thank you for  efforts  with us  I just wish to recommend  more detailed discussion  using the data from the results on the comment part  some of them are mentioned in the limitation part. 

Good luck  and well done 

Reviewer 2 Report

Dear Authors,

The piece entitled:

"Using Non-Pharmaceutical Interventions and High Isolation of Asymptomatic Carriers to Contain the Spread of SARS-CoV-2 in Nursing Homes  "

is a valuable one and might actually fill an important knowledge gap in current literature on pandemics effects in nursing homes.

Yet it might be substantially strengthened for its transnational comparability beyond the US health system.

For this purpose I warmly recommend that evidence base being cited is expanded and diversified to encircle far more sources referring to comparable pathways in other large nations particularly the wealthy emerging markets being the engine of world's economy recovery in post-Covid era.

Please consider some of these sources below alongside few additional ones at authors own disposal:

https://www.tandfonline.com/doi/full/10.1080/14737167.2020.1823221

https://www.frontiersin.org/articles/10.3389/fpubh.2021.673542/pdf

https://www.mdpi.com/1660-4601/17/24/9404/htm

https://www.ncbi.nlm.nih.gov/pmc/articles/PMC7553250/

https://openpublichealthjournal.com/VOLUME/13/PAGE/734/FULLTEXT/
